



# A Cavity-Enhanced UV Absorption Instrument for High Precision, Fast Time Response Ozone Measurements

Reem A. Hannun[1,2], Andrew K. Swanson[1,3], Steven A. Bailey[1], Thomas F. Hanisco[1], T. Paul Bui[4], Ilann Bourgeois[5,6], Jeff Peischl[5,6], Thomas B. Ryerson[5]

[1]Atmospheric Chemistry and Dynamics Laboratory, NASA Goddard Spaceflight Center, Greenbelt, MD, USA
[2]Joint Center for Earth Systems Technology, University of Maryland Baltimore County, Baltimore, MD, USA
[3]Universities Space Research Association, Columbia, MD, USA
[4]Earth Science Division, NASA Ames Research Center, Moffett Field, CA, USA
[5]NOAA Chemical Sciences Laboratory, Boulder, CO, USA
[6]Cooperative Institute for Research in Environmental Sciences, University of Colorado Boulder, Boulder, CO, USA

*Correspondence to*: Reem A. Hannun (reem.a.hannun@nasa.gov)

**Abstract.** The NASA Rapid Ozone Experiment (ROZE) is a broadband cavity-enhanced UV absorption instrument for the detection of *in situ* ozone ($O_3$). ROZE uses an incoherent LED light source coupled to a high-finesse optical cavity to achieve an effective pathlength of ~104 m. Due to its high-sensitivity and small optical cell volume, ROZE demonstrates a $1\sigma$ precision of 80

pptv (0.1 s) and 31 pptv (1 s), as well as a $1/e$ response time of 50 ms. ROZE can be operated in a range of field environments, including low- and high-altitude research aircraft, and is particularly suited to $O_3$ vertical flux measurements using the eddy covariance technique. ROZE was successfully integrated aboard the NASA DC-8 aircraft during July–September 2019 and validated against a well-established chemiluminescence measurement of $O_3$. A flight within the marine boundary layer also demonstrated flux measurement capabilities, and we observed a mean $O_3$ deposition velocity of $0.029 \pm 0.005$ cm s$^{-1}$ to the ocean

surface. The performance characteristics detailed below make ROZE a robust, versatile instrument for field measurements of $O_3$.

## 1    Introduction

In the troposphere, ozone ($O_3$) adversely affects air quality and acts as a greenhouse gas. Dry deposition to the Earth's terrestrial and oceanic surfaces represents a significant loss pathway for tropospheric $O_3$ (Young *et al.*, 2018) and thus influences tropospheric composition and $O_3$ pollution. Additionally, $O_3$ uptake through plant stomata leads to vegetation and crop damage (Ainsworth *et*

*al.*, 2012; Mills *et al.*, 2018) and poor ecosystem health (Lombardozzi *et al.*, 2015), potentially amplifying the effects of $O_3$ on climate (Sitch *et al.*, 2007) and air quality (Sadiq *et al.*, 2017). Despite its role in the tropospheric $O_3$ budget, dry deposition velocities ($v_d$) of $O_3$ remain poorly constrained (Wesely and Hicks, 2000; Hardacre *et al.*, 2015). The observational records of terrestrial $v_d(O_3)$ are limited in number and do not capture the full variability in $O_3$ deposition rates with land cover (Clifton *et al.*, 2020a). Furthermore, studies of $O_3$ deposition to the ocean (e.g., Kawa and Pearson, 1989; Faloona *et al.*, 2005; Helmig *et al.*,

2012; Novak *et al.*, 2020) report deposition velocities of ~ 0.01–0.05 cm s$^{-1}$, which are 1–2 orders of magnitude lower than typical terrestrial values. Observations from Helmig *et al.* (2012) and Novak *et al.* (2020) also suggest that $O_3$ deposition may vary with sea surface temperature. Global chemistry modeling frameworks that incorporate $O_3$ dry deposition (e.g., Bey *et al.*, 2001; Lamarque *et al.*, 2012) often apply fixed deposition rates to the ocean and heavily parameterized deposition schemes over land (Wesely, 1989). However, process-level representation of $O_3$ deposition improves agreement between modeled and observed

surface $O_3$ concentrations (Clifton *et al.*, 2020b; Pound *et al.*, 2020). The range and variability in $O_3$ deposition rates thus motivates the need for further $v_d(O_3)$ measurements to refine both atmospheric and land surface model predictions.





Measurements of vertical O₃ fluxes are typically accomplished via eddy covariance (EC) analysis. The EC technique demands fast time-response, high-precision sensors to resolve the turbulence-driven variability in scalar concentrations. O₃ fluxes are therefore

measured using highly sensitive O₃ detection methods such as chemiluminescence (e.g., Bariteau *et al.*, 2010; Muller *et al.*, 2010) and, more recently, chemical ionization mass spectrometry (CIMS) (Novak *et al.*, 2020). Chemiluminescence detectors employ either nitric oxide (NO) gas or organic dyes, which generate photons on reaction with O₃. While these instruments exhibit good sensitivity, they have practical drawbacks involving the use of toxic compressed gas cylinders and chemical dyes. Novak *et al.* (2020) successfully demonstrated the use of oxygen anion CIMS to measure O₃ and its vertical fluxes with a detection limit of <

0.005 cm s⁻¹ over the ocean. To the best of our knowledge, ultraviolet (UV) absorption instruments have not previously been utilized for O₃ flux measurements due to insufficient sensitivity (e.g., Gao *et al.*, 2012). However, advancements in incoherent cavity-enhanced absorption spectroscopy (Fiedler *et al.*, 2003) facilitate the development of high-sensitivity sensors that are both robust and compact. Furthermore, UV absorption has the advantage of providing direct detection of O₃ without the need for a chemical titration source.


We report on the development of the NASA Rapid Ozone Experiment (ROZE), a cavity-enhanced UV absorption instrument for the *in-situ* detection of O₃. The long optical pathlength and small cavity volume enable high precision measurements in short averaging times, making ROZE suitable for O₃ flux measurements with the EC technique. The compact instrument design supports integration aboard research aircraft for both tropospheric and stratospheric deployment. We describe the principle of operation

along with major instrument components and performance characteristics below. We also discuss the field performance of ROZE and demonstrate its EC capabilities using aircraft observations of O₃ deposition to the ocean surface.

## 2    Principle of operation

Incoherent broadband cavity-enhanced absorption spectroscopy (IBBCEAS) is an established tool for the detection of trace gas species (Fiedler *et al.*, 2003; Ball *et al.*, 2004; Washenfelder *et al.*, 2008) including O₃ (Darby *et al.*, 2012; Gomez and Rosen,

2013). IBBCEAS relies on a broadband, incoherent light source coupled to a high-finesse optical cavity. Typically, a multi-channel detector resolves structured absorption features in the ultraviolet (UV) or visible spectral regions. IBBCEAS exploits the long optical pathlength generated in the cavity to enhance sensitivity, comparable to other cavity-enhanced methods such as cavity ringdown spectroscopy (CRDS). However, unlike CRDS, IBBCEAS uses a relatively inexpensive light source as compared to a narrow linewidth laser. Furthermore, the incoherent light source relaxes the stringent requirements for cavity alignment that

accompany other cavity enhanced methods such as CRDS, enabling a more robust instrument configuration for field environments.

ROZE employs the IBBCEAS technique for high-sensitivity measurements of O₃. As illustrated in Figure 1, a light-emitting diode (LED) in the UV ($\lambda_{\max}$ = 265 nm) is collimated and coupled into an optical cavity via high-reflectivity mirrors. Exiting light is passed to a photomultiplier tube (PMT) detector through a series of collection and filter optics. Figure 2 depicts the normalized

detected LED intensity, which accounts for the LED spectral irradiance, the optical bandpass filter transmission, and the wavelength dependent PMT response. The LED spectrum overlaps with the O₃ Hartley band, and any O₃ present in the sample cell attenuates the light intensity received at the detector. The use of optical filters on the PMT precludes the need for wavelength resolution from a grating spectrometer and simplifies data reduction. Section 3.1 provides further details on the optical system.



Attenuation of light intensity in an IBBCEAS cavity results from trace gas absorption as well as extinction due to the mirrors and Rayleigh scatter. Accounting for these additional losses, the Beer-Lambert absorption coefficient, $\alpha_{abs}$, is related to the observed change in intensity transmitted through the cavity as follows (Washenfelder *et al.*, 2008):

$$\alpha_{abs} = \left(\frac{I_0 - I}{I}\right)\left(\frac{1-R}{d} + \alpha_{Ray}\right) \tag{1}$$

Here, $I_0$ is light intensity in the absence of any absorbing species, $I$ is the intensity attenuated due to absorption, $R$ is the mirror

reflectivity, $d$ is the physical distance separating the cavity mirrors, and $\alpha_{Ray}$ is the extinction due to Rayleigh scatter, a non-negligible component in the UV. The term $(1 - R)/d$ gives the theoretical cavity loss, $\alpha_{cav}$, and represents the inverse of the maximum effective pathlength, $L_{\text{eff}}$. In cavity-enhanced techniques, $L_{\text{eff}}$ can be many orders of magnitude larger than $d$, resulting in high sensitivity to the absorbing species. Equation 1 can also be expressed as $\alpha_{abs} = N\sigma_{abs}$, where $N$ is number density of the absorbing species and $\sigma_{abs}$ is the absorption cross section. In principle, accurate trace gas measurements require calibration of the

$\alpha_{cav}$ term, as well as knowledge of the Rayleigh and absorption cross sections in the detected spectral region. The data processing and calibration for ROZE will be discussed in Sections 3.4 and 4.1, respectively.

## 3    Instrument description

ROZE consists of three main subsystems housed in a compact 58 cm long x 44 cm wide x 18 cm high chassis, with a total instrument weight of 19 kg (Figure 3). The optical plate – a custom aluminum honeycomb panel supported by friction dampened spring

vibration isolators – provides a stable platform for the optical components, consisting of the LED, sample cell, and PMT. The remaining subsystems include the flow handling and the data acquisition. Each major subsystem is described in greater detail below. ROZE operates at 24 $V_{DC}$ with a low-profile AC-DC switching power supply (Vicor VI-LU3-IU) capable of running off 115 or 230 $V_{AC}$ (47–440 Hz), which can be supplied directly from the aircraft. Power consumption is less than 200 W and typically ~100 W. Table 1 summarizes ROZE design and performance characteristics.

### 3.1    Optical components

#### 3.1.1    LED assembly

A UV LED ($\lambda_{\text{max}}$ = 265 nm, FWHM = 10 nm) (Thorlabs M265D2) is mounted to a custom heat sink and temperature controlled to 30 °C with a thermo-electric cooler (TE Technology CH-21-1.0-1.3; Wavelength Electronics PTC2.5K-CH). The LED output power is separately monitored by a photodiode (Marktech MTPD4400D-1.5) inserted into the edge of a lens tube that holds the

LED. The LED assembly attaches to a custom cage mount system that also houses the associated optics, including an aspheric collimation lens ($f$ = 79 mm, Thorlabs ASL10142M) and a beam expander (Thorlabs BE02-UVB) in reverse to shrink the collimated LED output. For compactness, the LED assembly and cage system are mounted parallel to the sample cell, and two mirrors (Thorlabs NB1-K04) direct the beam 180° into the cell.

#### 3.1.2    Sample cell

The sample cell is manufactured from an aluminum alloy tube measuring 30 cm in length with a 1.2 cm inner diameter. The cell mirrors (Layertech 109561) have a reflectivity of $R$ > 99.7% over the detected spectral range (Figure 2) and a 500 mm radius of curvature. The mirrors are held directly at the cell ends on face type o-ring seals using custom, non-adjustable mounts fastened to tube collars. The mirror positions are configured to maximize centricity. Two gas ports direct the sample flow into and out of the cell at right angles. The sample enters through a custom cylindrical diffuser, a ring with circumferential openings adjacent to the





cell mirrors, that nests within the cell tube orthogonal to the ports. The diffuser helps minimize noise due to Rayleigh scatter from turbulence within the cell at high sample flow rates. A pressure transducer (Omega MMA015V10P4K1T4A6) measures the cell pressure from a port near the cell center. The entire cell is thermally regulated to 35 °C using resistive heaters and a precision heater control (Wavelength Electronic PTC2.5K-CH).

### 3.1.3 PMT assembly

A PMT (Hamamatsu H10720-113) operating in analog mode collects the light exiting the cell. Two optical bandpass filters (Thorlabs FGUV5-UV, Semrock FF01-260/16) transmit the cell output to a collection lens ($f$ = 35 mm, Thorlabs LA4052-UV), which images the beam onto the PMT photocathode. A UV window (Thorlabs WG40530-UV) glued into a custom PEEK lens tube adapter seals to the PMT face with a Viton gasket, creating a leak-tight package for low-pressure (high-altitude) operation. The PMT is thermally stabilized to 35 °C in the same manner as the sample cell. The PMT signal is passed to an amplifier circuit

(Analog Devices EVAL-ADA4625-1ARDZ) before digitization by the data acquisition system described below.

### 3.2 Flow system

The ROZE flow system is designed to achieve rapid flushing of the detection cell as required for fast concentration measurements. However, ROZE samples at ambient pressure to maximize sensitivity, necessitating high throughput with a minimal pressure differential. ROZE utilizes a linear diaphragm pump (Thomas 6025SE-150113) that can achieve a flow rate of up to 18 standard

liters per minute (SLM) through the system. The pump speed can also be adjusted by varying the supply current and has three pre-set speeds that can be changed by a switch on the chassis front panel. A flow meter (Honeywell AWM5104) located between the cell exhaust and the pump monitors the sample flow in real time.

ROZE O$_3$ measurements also require knowledge of the reference intensity ($I_0$) as detailed in Equation 1. A 3-way solenoid valve

(NResearch TC648T032) switches between the sample line (ambient air from the aircraft inlet) and the zero port, which attaches to an internal Carulite O$_3$ scrubber (2B Technologies) to produce O$_3$-free air. Periodic zeroing during operation captures long-term drift in $I_0$ due to the LED output, PMT response, and changing environmental conditions. Typically, the instrument opens to the O$_3$ scrubber for 10 seconds every 5 minutes.

ROZE uses fluorinated ethylene propylene (FEP) tubing both external and internal to the chassis upstream of the sample cell. External to the chassis, the inlet details depend on the aircraft platform. ROZE has previously used the inlet detailed in Cazorla *et al.* (2015) when flying on the NASA DC-8 aircraft. The instrument exhaust plumbs directly to an exhaust port near the rear of the aircraft. To exclude dust and other particles from affecting the mirror reflectivity, a 2-micron pleated mesh filter (Swagelock) affixes to the sample cell inlet port. More aggressive filtering can be achieved at the cost of reduced (< 18 SLM) flow rates but

may be necessary depending on the environment and if O$_3$ deposition measurements are not the primary goal.

### 3.3 Data acquisition

ROZE utilizes a CompactRIO (National Instruments cRIO-9030) that incorporates a real-time operating system and a Field Programmable Gate Array (FPGA). The FPGA is configured for modulation of the LED and subsequent digitization of the PMT signal. To improve measurement precision and remove background due to ambient light scatter, the FPGA modulates the LED at

1 kHz with a 90% duty cycle via an external LED driver (Wavelength Electronics FL591FL). A 16-bit Analog to Digital Converter (ADC) digitizes the amplified PMT signal at a digitization rate of 100 kHz. This high rate enables us to average each LED ON and



OFF pulse amplitude. We then take the difference of the ON and OFF signals to remove background noise, both optical (i.e., stray light) and electronic. The 1 kHz differences are further averaged to 10 Hz and recorded. Other diagnostic housekeeping variables (e.g., sample flow, temperatures, LED power) are recorded at 1 Hz. Additionally, an analog output commands the 3-way valve to

open to the zero line with a user-defined period and duration.

### 3.4 Data processing

In practice, the absorbance calculation for ROZE factors in the pressure difference between the sample and zero lines, as derived by Min *et al.* (2016):

$$\alpha_{O3} = \left(\frac{I_Z}{I} - 1\right)\left(\alpha_{cav} + \alpha_{Ray,Z}\right) + \Delta\alpha_{Ray} \qquad (2)$$

Analogous to Equation 1, $I_Z$ is the intensity measured when sampling through the zero line ($O_3$-scrubbed air), $I$ is the intensity when sampling ambient air, and $\Delta\alpha_{Ray} = \alpha_{Ray,Z} - \alpha_{Ray,S}$, where $\alpha_{Ray,Z}$ and $\alpha_{Ray,S}$ give the Rayleigh extinction ($\alpha_{Ray} = N_{air}\sigma_{Ray}$) of the zero and the sample respectively. The $O_3$ number density can then be determined as $\alpha_{O3} = N_{O3}\sigma_{O3}$. The Rayleigh scattering (Bucholtz, 1995) and $O_3$ absorption (Serdyuchenko *et al.*, 2014) cross sections are calculated as the weighted average over the collected spectral range (Figure 2). Using known cross sections and a calibrated $\alpha_{cav}$ (inverse effective pathlength), the

observed change in intensity yields a direct measure of the $O_3$ concentration.

## 4 Performance

### 4.1 Sensitivity and calibration

The effective pathlength of the ROZE optical cavity determines the instrument sensitivity to $O_3$ (i.e., the attenuation in intensity per unit $O_3$). The cavity extinction, and thus the effective pathlength, are dictated by the mirror reflectivity as described above but

require independent calibration. Calibration can be accomplished via standard addition of $O_3$ or Rayleigh attenuation (in the absence of absorbing species) at varied sample pressures. The former method relies on commercially available $O_3$ generators or sensors for verification, which lack the required accuracy and may drift over time. In contrast, the Rayleigh calibration provides a convenient and straightforward alternative. Both methods are described below.

Figure 4a depicts the ROZE calibration using known concentrations of $O_3$. A commercial $O_3$ source (2B Technologies 306) generated known amounts of $O_3$, with the zero $O_3$ addition serving as the $I_Z$ baseline. Per Equation 2, the slope of the observed attenuation ($dI = I_Z/I - 1$) as a function of $O_3$ number density is proportional to the remaining extinction terms ($\alpha_{cav} + \alpha_{Ray}$). Solving for $\alpha_{cav}$ using the $O_3$ cross section and the calculated Rayleigh extinction, the calibration yields an effective pathlength of $L_{eff} = 108 \pm 6$ m. The alternate calibration uses the Rayleigh extinction in zero air over a range of cell pressures (Figure 4b). In

the absence of absorbing species, an expression for $\alpha_{cav}$ can be derived following the approach in Washenfelder *et al.* (2008) as:

$$\alpha_{Ray} = \left(\frac{I_0}{I} - 1\right)\alpha_{cav} \qquad (3)$$

$I_0$ represents the intensity at vacuum, which can be extrapolated from a linear fit of counts as a function of cell pressure. The slope of the observed change in intensity with number density therefore yields a direct measure of the cavity extinction, resulting in an effective pathlength of $104 \pm 4$ m. The two methods agree to within fit uncertainties, and we use $L_{eff}$ as determined by the Rayleigh

calibration for subsequent calculations.



### 4.2     Precision and accuracy

The major contributions to instrument noise include PMT electrical noise and differential scatter or absorption due to non-uniform flow within the sample cell at high flow rates. The flow diffuser (see Section 3.1.2) effectively reduces the flow noise, while decreasing the gain on the PMT amplifier circuit minimizes the PMT electrical noise. The ROZE precision can be determined from the continuous sampling of zero air at a constant pressure. Figure 5 depicts a $1\sigma$ Allan deviation plot for ROZE (in pptv $O_3$ equivalents) as calculated from optical extinction measurements of zero air acquired over 1.5 hours at 944 mbar. For short integration times ($< 10$ s), a fit of the data gives a $\tau^{-0.47}$ decay, indicating the Allan deviation closely follows the square root of the averaging time ($\tau^{-1/2}$) as expected for white noise.  At the native 0.1 s sampling rate, the $1\sigma$ precision for $O_3$ is 80 pptv and reduces to 31 pptv with 1 s averaging. For the given cell pressure and a temperature of 35 °C, this translates to a $1\sigma$ precision of $6.7 \times 10_8$ molecules cm$_{-3}$ (1 s average) of $O_3$.

The absolute accuracy of the ROZE measurement depends on uncertainties in the literature-reported values of the $O_3$ and Rayleigh cross sections, the measured cell temperature and pressure, and the calibrated cavity extinction. The reported $O_3$ absorption cross section has an uncertainty of 2% (Gorshelev *et al.*, 2014), and we estimate an upper uncertainty of 3% for the Rayleigh scattering cross section (Bucholtz, 1995). The cell pressure and temperature are accurate to within 0.2% and 0.5% respectively, and the calibrated cavity extinction has an additional 4% slope uncertainty from the linear fit. These errors propagate through Equation 2 to yield a total measurement uncertainty of 6.2% in the $O_3$ number density.

### 4.3     Response time

The flush time of the sample cell limits the true instrument response time despite the 10 Hz data acquisition rate. A rapid flush rate is critical for high spatial resolution measurements from a fast-moving platform. Additionally, fast concentration measurements are required for sampling of turbulent eddies for airborne EC, and the necessary time response scales with aircraft speed. Response times of 10 Hz are typically considered sufficient for ground-based EC (Aubinet *et al.*, 2012), while for airborne EC, a response times of 1–5 Hz are typically sufficient due to larger eddy scales at altitude (Wolfe *et al.*, 2018). Figure 6a shows the instantaneous instrument response to a series of 10 ms pulses of $O_3$ injected into a zero-air carrier flow using a fast switching valve (The Lee Company, IEP series). During this experiment, the pump maintained a sample flow rate of 18 SLM. A series of exponential decay fits for several $O_3$ pulses yields an *e*-folding time constant of $\tau_r = 50 \pm 4$ ms (Figure 6b). The time constant corresponds to a $3e$-fold flush rate of 9.5 Hz.

### 5     Field demonstration

ROZE can be operated on both low- and high-altitude aircraft platforms. Though ROZE has not yet flown on a high-altitude unpressurized aircraft (such as the NASA ER-2), laboratory experiments in a thermal vacuum chamber have demonstrated no loss of performance down to a pressure and temperature of 50 mbar and 250 K (results not shown).

In summer 2019, ROZE flew aboard the NASA DC-8 for the Fire Influence on Regional to Global Environments Experiment, Air Quality (FIREX-AQ) campaign over the Central and Northwest United States. Campaign flights targeted smoke plumes from forest wildfires and agricultural burns. The instrument operated as described above, with the addition of an inline particle filter (Balston 9922-05-DQ) to protect the cavity mirrors from fine particulates in the smoke. In fresh, concentrated smoke plumes, UV-active species such as $SO_2$ and aromatic hydrocarbons can give rise to positive artifacts in the $O_3$ absorption measurement (Birks,

2015). This is a drawback to the UV absorption measurement, which can be affected by both absorbing and scattering molecules. However, such UV-active absorbers are generally not abundant enough in the background atmosphere to be of concern. The

FIREX-AQ ROZE data are therefore quality filtered to remove points sampled within dense smoke plumes using observed formaldehyde mixing ratios above 5 ppbv. Below, we detail comparisons of ROZE against an established $O_3$ measurement. Additionally, level flight legs in the marine boundary layer during a flight over the ocean provide an initial demonstration of $O_3$ vertical flux measurements.

### 5.1 Validation against chemiluminescence

The DC-8 FIREX-AQ payload included the NOAA Nitrogen Oxides and Ozone (NOyO3) instrument, a well-established $O_3$ measurement using the chemiluminescence technique (Ryerson *et al.*, 2000; Bourgeois *et al.*, 2020). ROZE operated simultaneously with the NOyO3 instrument during several flights. Figure 7 shows a comparison of ROZE and NOyO3 data for the July 30, 2019 flight over the Northwestern United States. During this flight, no fresh smoke plumes were sampled, and no filtering of the ROZE data was necessary. Figure 7a depicts a ~25 min subset of the full timeseries to illustrate the ROZE instrument

precision. Both measurements (averaged to 1 s) track the dynamic features in $O_3$ mixing ratios well. The cross plot for the full flight (Figure 7b) demonstrates strong agreement between the two measurements, with a slope of $0.98 \pm 0.01$ and an intercept of $0.17 \pm 0.02$ ppbv $O_3$ ($r_2 = 0.99$). Comparisons for 15 flights from the campaign indicate a range in slopes of 0.96–1.04, consistent with the measurement uncertainty.

### 5.2 Ozone flux measurements

#### 5.2.1 Eddy covariance flux

The vertical flux of $O_3$ can be directly quantified using the eddy covariance (EC) technique. EC defines the flux ($F$) as the time or spatially averaged covariances in the vertical wind speed ($w$) and the scalar species of interest (in this case the $O_3$ mixing ratio $X_{O3}$):

$$F_{O3} = \langle w' X_{O3}' \rangle \tag{4}$$

In the equation above, the primes denote instantaneous deviations from the mean value, and the brackets indicate an average over a prescribed interval as discussed below. Since deposition dominates transfer across the air-surface interface, the $O_3$ flux can instead be expressed as a transfer rate or deposition velocity ($v_d$):

$$v_d = -\frac{F_{O3}}{\overline{X}_{O3}} \tag{5}$$

Here, the overbar indicates the mean $O_3$ mixing ratio over the averaging period. The deposition velocity, in units of cm s$_{-1}$, yields

a normalized metric of the deposition efficiency and incorporates both chemical and physical transfer processes.

During the FIREX-AQ campaign, the flight on July 17, 2019 contained a level segment within the turbulent marine boundary layer suitable for EC. The flux transects were located over the Pacific Ocean, ~200 miles southwest of the Los Angeles Basin. To quantify $O_3$ deposition, the Meteorological Measurement System (MMS) instrument provided 3-D wind vector data (Chan *et al.*,

1998), which were used in conjunction with ROZE $O_3$ measurements. A 1-D coordinate rotation was applied to the wind vector to force the mean vertical wind to zero, and the native 20 Hz MMS data was averaged to the ROZE 10 Hz time base. Note that the additional particle filter reduced the ROZE sample flow to 11.3 SLM, and we estimate the time constant from the decay of zero-$O_3$ additions as $\tau_r = 90$ ms (5.5 Hz 1/3$e$ flush rate). We also use 20 Hz water vapor measurements from the open path Diode Laser Hygrometer (DLH) (Diskin *et al.*, 2002) as a benchmark for the flux performance. 20 Hz DLH data were averaged to the ROZE





time base and used to apply a moist-to-dry air correction for raw $O_3$ observations, negating the need for density corrections to the calculated flux (Webb *et al.*, 1980). This density correction reduces the $O_3$ flux by ~6%. For the EC calculations, we selected two ~50 km transects with consistent aircraft heading, stationary flow, and level altitude (~170 m). Scalar data processing included detrending the scalar mixing ratios by subtracting a 20 second running mean and synchronizing the data with the vertical winds.

### 5.2.2    Spectral analysis

Spectral analysis aids in decomposing the contributions of eddies at different scales (frequencies) to the overall signal and provides a quality assessment of the ROZE flux measurements. Figure 8 displays the lag-covariance, power spectrum, and co-spectrum for $O_3$ and vertical wind fluctuations generated using fast Fourier transforms (FFTs) for a single transect. The spectra for water vapor are also displayed for comparison. The lagged cross-cross covariance functions (Figure 8a) demonstrate defined peaks at lags of < 0.5 s, with the peak non-normalized covariance yielding a measure of the flux. Dividing out the background $O_3$ mixing ratio of 29 ppbv, we find a mean deposition velocity of 0.029 cm s–1 for the two transects. The power spectra in Figure 8b show that vertical winds follow the theoretical $f^{-5/3}$ decay expected in the inertial subrange (Kaimal *et al.*, 1972). The slope for the $O_3$ power spectrum initially follows the same decay but flattens at ~1 Hz, indicating that the turbulence-driven variability in $O_3$ approaches the ROZE precision limit in higher-frequency eddies. However, the normalized frequency-weighted co-spectral power of $w'$ with $X_{O_3}'$ (Figure 8c, solid lines) show that flux carrying eddies below ~0.6 Hz dominate the total signal. The ogive, the cumulative integral of the co-spectrum (Fig. 8c, dashed lines) further indicates that 99% of flux carrying eddies occur at frequencies below ~4 Hz. These results demonstrate the adequate ROZE time response for airborne EC.

### 5.2.3    Flux uncertainty

Detailed methods to quantify flux errors for airborne EC can be found elsewhere (Lenschow *et al.*, 1994; Langford *et al.*, 2015; Wolfe *et al.*, 2018). Here, we aim to quantify the random and systematic flux errors that reflect the overall instrument performance. We use the empirical formulation of Finkelstein and Sims (2001) to estimate the total random error ($RE_{TOT}$) as the variance of the scalar-wind covariance. In this approach, the $RE_{TOT}$ is determined using auto- and cross-correlation functions (as in Figure 8a) over lag times that are sufficient to capture the timescale of the correlation (here ~10 s). Averaging over the flux legs yields a $RE_{TOT}$ of 0.005 cm s–1. $RE_{TOT}$ encompasses both instrument noise as well as error from the random sampling of turbulence. To isolate the RE component due solely to instrument noise ($RE_{noise}$), we follow the approach of Mauder *et al.* (2013). In this method, the standard deviation of the instrument noise is derived from the scalar auto-covariance and then propagated to determine its contribution to the to the cross-covariance uncertainty. Note that $RE_{noise}$ still depends on the turbulence regime and therefore varies with atmospheric conditions. We calculate $RE_{noise}$ to be 0.0015 cm s–1 averaging over the two flux transects. These results indicate that instrument noise constitutes ~30% of the total random error.


Additionally, the instrument time response can lead to systematic flux errors as a consequence of under sampling contributions from high-frequency eddies. We determine the systematic error due to the instrument response time ($SE_{RT}$) following the Horst (1997) model, whereby the attenuation in the measured signal can be expressed as a co-spectral transfer function based on the characteristic instrument response time. Using the ROZE response time of $\tau_r$ = 90 ms, we determine $SE_{RT}$ as < 2%, indicating minimal attenuation in the measured flux signal.



## 6    Summary and conclusions

The NASA ROZE instrument provides high sensitivity, fast time response measurements of $O_3$ via broadband cavity-enhanced UV absorption. The compact, robust instrument package is adaptable to diverse field environments, including low- and high-altitude aircraft platforms. ROZE currently achieves a $1\sigma$ precision of ~30 pptv $s_{-1}$ and an overall accuracy of 6.2%. ROZE was

successfully integrated aboard the NASA DC-8 aircraft, and the field performance compares favorably with an independent $O_3$ measurement to within ROZE uncertainty. The maximum observed time response for laboratory tests was 50 ms, with additional filtering during aircraft operation slowing the time response to 90 ms. The instrument precision and time response make ROZE particularly well suited for vertical $O_3$ flux measurements using eddy covariance analysis. ROZE has measured $O_3$ deposition velocities of $0.029 \pm 0.005$ cm $s_{-1}$ to the ocean surface, with minimal (< 2%) response-time attenuation in the flux signal. The

demonstrated performance of ROZE makes the instrument an ideal and versatile option for field measurements of both $O_3$ concentrations and fluxes.

### Data availability

The FIREX-AQ data for $O_3$ (ROZE and NOyO3), water vapor (DLH), formaldehyde (ISAF), and 3-D winds (MMS) are publicly available at https://www-air.larc.nasa.gov/missions/firex-aq/.

### Acknowledgements

This work was supported by the NASA Internal Research and Development (IRAD) program, the NASA Upper Atmosphere Research Program, and the NASA Tropospheric Chemistry Program. The aircraft flight opportunity was provided by the NASA/NOAA FIREX-AQ project and the NASA Student Airborne Research Program (SARP). We would like to acknowledge the DLH instrument team (Glenn Diskin, et al.) for the water vapor measurements used in the eddy covariance analysis. We would

additionally like to thank Jason St. Clair and Glenn Wolfe for helpful comments on the manuscript.

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



**Table 1: Summary of ROZE performance capabilities.**

| Specification | Value |
| --- | --- |
| Size | 58 x 44 x 18 cm |
| Weight | 19 kg |
| Power | < 200 W |
| Data rate | 10 Hz |
| Precision ($1\sigma$, 1Hz) | 6.7 x $10_8$ molec. cm$_{-3}$ |
| Accuracy | 6.2% |
| Time response | 50 ms |




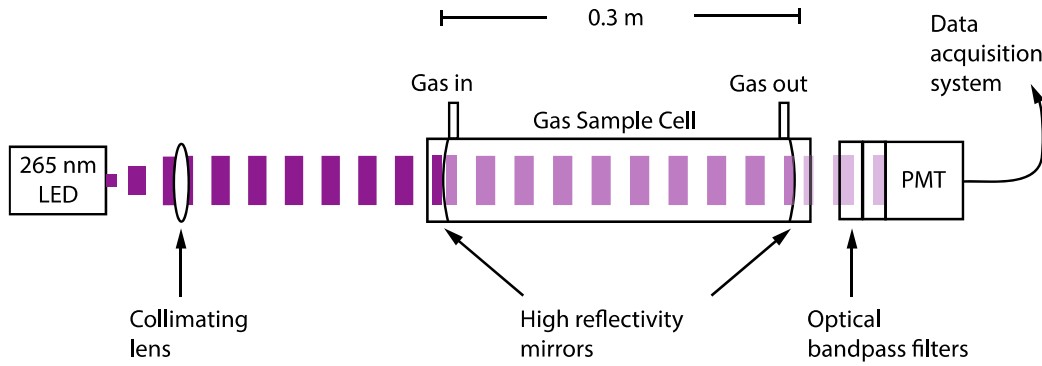

**Figure 1: Incoherent broadband cavity enhanced detection technique for O₃. A LED at 265 nm is collimated and coupled into the detection cell via high reflectivity mirrors (R > 99.7%), creating a long optical pathlength. The light attenuated by the sample is then detected using a photomultiplier tube (PMT) operated in analog mode. The sample enters and exits the cell orthogonal to the beam propagation.**


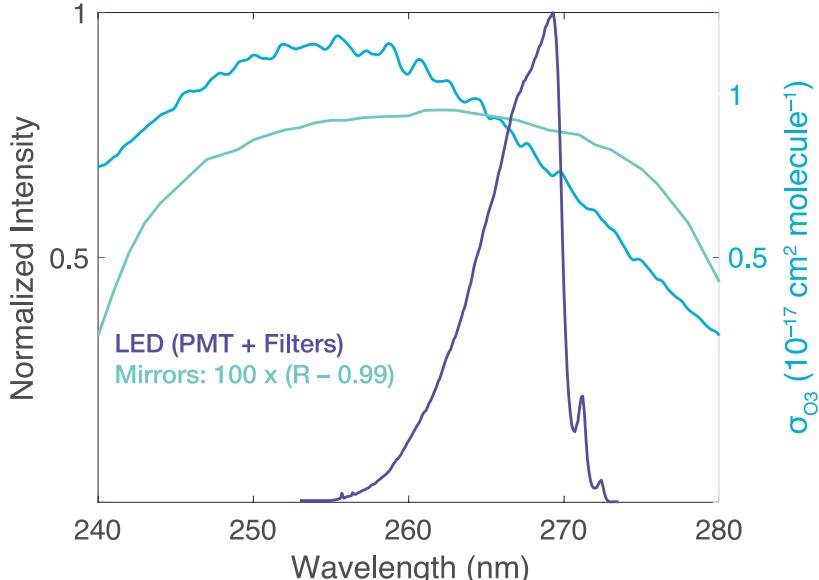

**Figure 2: LED spectrum, mirror reflectivity, and O₃ absorption cross section: The LED ($\lambda_{max}$ = 265 nm, FWHM = 10 nm) spectrum was measured using a grating spectrometer (0.1 nm resolution) with the instrument PMT and associated detector optics. The mirror curve depicts $100 \times [R - 0.99]$, where $R$ is the reflectivity, over a range of wavelengths. The right axis shows the absorption cross section for the O₃ Hartley band. O₃ and Rayleigh cross sections were determined as the weighted average with the normalized intensity of the LED and PMT detector optics.**




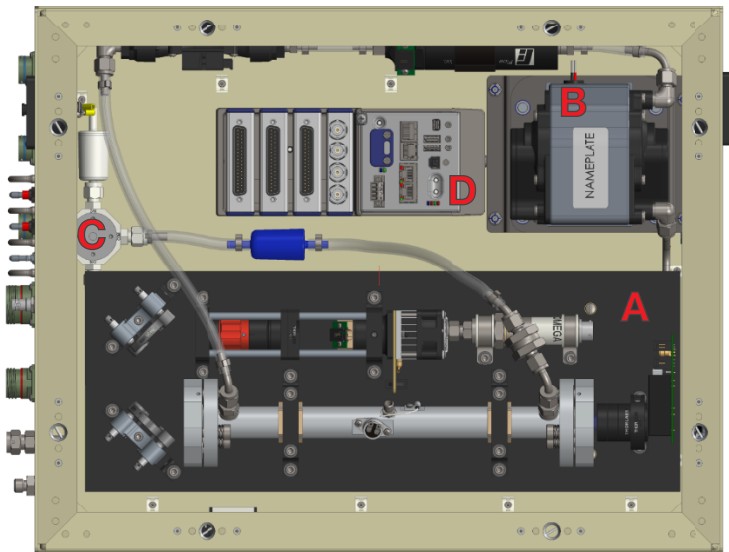

**Figure 3: A top view of the ROZE instrument chassis. Major components include A) the optical plate, which consists of the LED assembly, associated optics, the optical cell, and PMT detector; B) The diaphragm pump which can pull up to ~18 SLM through the flow system; C) The 3-way valve which switches between the sample line and air scrubbed of $O_3$ using a Carulite filter; and D) The data acquisition system.**



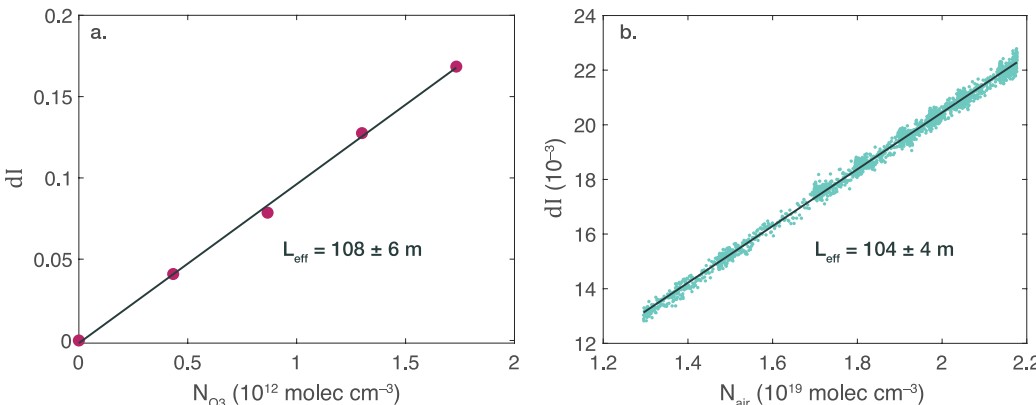


**Figure 4: ROZE calibration: a) The effective pathlength ($L_{eff}$) as determined by attenuation ($dI$) due to known additions of O₃ from a commercial ozone-generator. The slope yields the effective pathlength as determined from Equation 1 in the text using the known O₃ absorption cross section; b) Attenuation due to Rayleigh scatter over a range of cell pressures. The slope of attenuation as a function of number density gives the pathlength using the known Rayleigh scattering cross section for zero air. The pathlength derived from both calibrations agrees to within the fit uncertainty.**




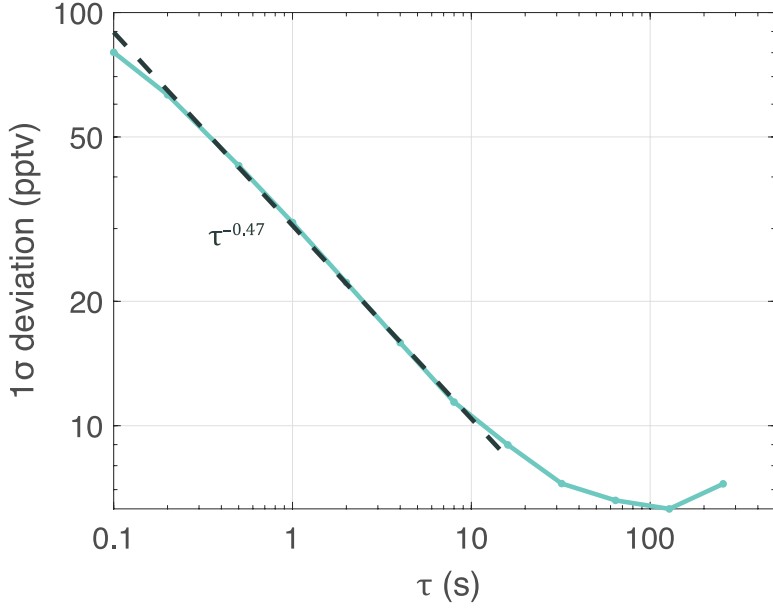


**Figure 5: Allan deviation plot for 1.5 hr of sampling zero air at constant pressure (944 mbar). The $1\sigma$ precision is expressed in pptv equivalents of $O_3$ as a function of the integration time $\tau$. The curve demonstrates a precision of 31 pptv in a 1 s integration time. The dashed line shows a $\tau^{-0.47}$ decay for short integration times (< 10 s), comparable to the $\tau^{-1/2}$ decay expected for white noise.**



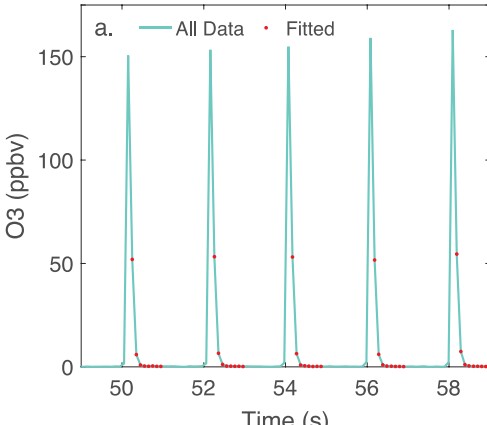
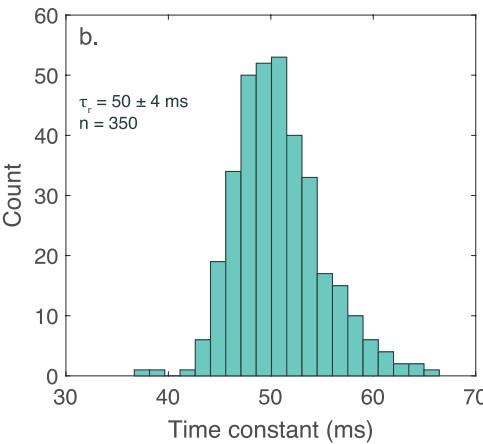

**Figure 6: ROZE time response: a) Ozone was injected into the flow system via a pulsed valve at 2-second intervals with a sample flow of 18 SLM. Individual pulses were fit to an exponential decay using the selected data points in red; b) Histogram of time constants for all 350 pulses. The *e*-folding decay time of 50 ± 4 ms corresponds to a (1/3*e*) flush rate of 9.5 Hz.**





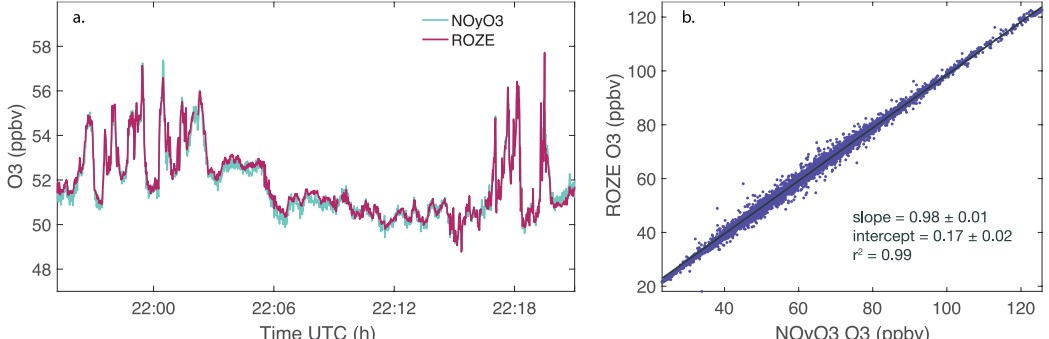

**Figure 7: ROZE and NOyO3 measurements of O₃ from a FIREX-AQ flight on July 30, 2019 over the Northwestern US: a)**
**Timeseries of ROZE and NOyO3 data (averaged to 1 s); b) Scatter plot of ROZE and NOyO3 O₃ measurements from the full flight. A linear fit to the data yields a slope of 0.98 ± .01 and an intercept of 0.17 ± 0.02 ppbv.**





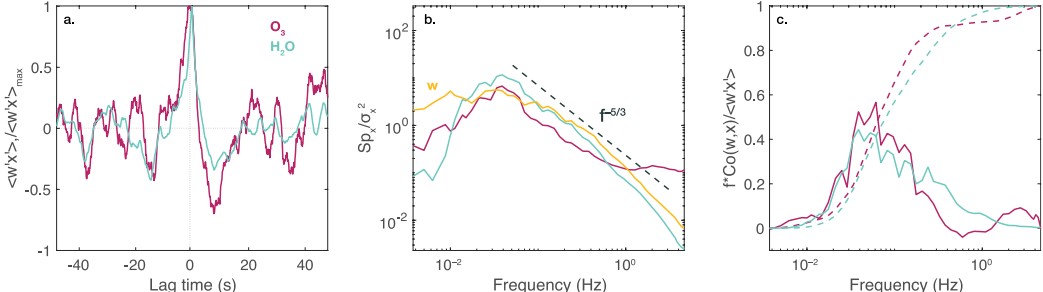


**Figure 8: Example spectra from a 50 km flux leg at 170 m altitude during the July 17, 2019 flight over the Pacific Ocean: a) Vertical wind-scalar ($w$ and $x$ respectively) cross covariance functions normalized by the maximum covariance for $O_3$ and water vapor; b) Power spectra normalized to total variance for $w$, $O_3$, and $H_2O$. The dashed line represents the $f^{-5/3}$ theoretical decay for the inertial subrange; c) Solid lines depict co-spectral power (frequency-multiplied and covariance-normalized) of $O_3$ and $H_2O$ with vertical wind. Dashed lines depict the respective ogives (cumulative integrals).**
