# Peer review of "A Cavity-Enhanced UV Absorption Instrument for High Precision, Fast Time Response Ozone Measurements"

_Atmospheric Measurement Techniques, 2020_

## Referee Comment (RC1) · Anonymous Referee #1 · 10 Aug 2020

A Cavity-Enhanced UV Absorption Instrument for High Precision, Fast Time Response Ozone Measurements

Reem A. Hannun et al.

amt-2020-195

**General comments:**

This well-written and well-structured manuscript describes the development, validation and application of an instrument based on incoherent broadband cavity-enhanced absorption spectroscopy (IBBCEAS) for the direct detection and quantification of ozone in the UV-C region (~265 nm). The pulsed instrument (Rapid Ozone Experiment, ROZE) is designed for $O_3$ Eddie covariance measurements from airborne platforms and thus requires high time resolution. The authors give an excellent motivation for their work in the introduction. They briefly review the measurement principle before outlining the instruments operation, where sufficient attention to detail is provided to understand the function of the instrument and how the key objectives in instrument performance were met. The lab characterization of the instrument's performance addresses all relevant aspects. Finally the first application in airborne operation also shows the quality of the instrument for Eddie covariance measurements in comparison with a more established chemiluminescence (reference) approach. The authors pay sufficient attention to detail, error discussion, and relevant critical measurement parameters.
The manuscript is basically publishable in its current form, however subject to some improvement and amendments concerning more explanations as outlined below.

**Specific comments:**

L49: The UV region has the drawback that in this region many other trace species also have significant absorption bands and this puts substantial constraints on the selectivity of the method in that region. See also comment below (L216-220).

L70-74: The use of an optical filter simplifies the approach and is adequate because the spectrum is unstructured and the absorption has not much or very little "fingerprint character". However, it also illustrates the low selectivity for ozone in this region (see L216-220).

L101: The divergence of the LED is quite large – the surface area of LED is also large 1x1 mm^2 (extended light source). Light collection at a distance of 79 mm with a 1" diameter optic causes substantial light loss. Using a beam expander in revers increases the divergence after partial collimation again by a factor of 2. Can the authors give an estimate what fraction of the overall emission from the LED is actually being imaged onto the cavity – in other words, what is the light collection efficiency for the LED?

L103: "…*direct the beam 180° into the cell.*"    This sounds strange. Please rephrase or at least refer to Figure 3 here.

L105-114 (section 3.1.2): There is no mention of a purge system. Were the cavity mirrors purged? If not – why not? There is also no mention of an aerosol filter in the inlet at this point, but later the authors point this out (since scattering in the UV is substantial). This section would benefit from mentioning these elements explicitly here.

L125/126: At this point I was wondering whether there was a flow controller? The flow seems to be only controlled by pump power? How accurate is the flow and how does it vary with pressure

variation? (Relevant for aircraft measurements on the ascent and descent). What are the three flows according to the 3 flow speed adjustments?

L145: Here it would be nice to explicitly learn what on- and off-times a 90% duty cycle refers to? Are there are delay times. Inherently pulsed IBBCEAS is not commonly applied and in the literature the large majority of instruments is indeed continuous wave. Pulsed IBBCEAS can in principle be biased by offsets depending on the measurement timing, which in turn may lead to systematic deviations in the measured absorption signal (see e.g. Keary & Ruth, Opt. Express, 2019). The authors may want to make an argument that these offsets are not observed here based on the electronics' timing. See also comment on Fig. 1 below.

L182/183: "…and differential scatter or absorption due to non-uniform flow within the sample cell at high flow rates." As in the comment above, more on flow variation and or flow control would be helpful. The instrument might benefit from a pressure controlling flow controller.

L205: "During this experiment, the pump maintained a sample flow rate of 18 SLM." Why was this not done at a lower mass flow of 11 SLM relevant for the airborne measurements with particle filter?

L206: I am a bit confused here: Please explain "3e fold flush rate". Also compare with caption of Fig 6 which says 1/3e is 9.5 Hz.  The 3e-folding time is 150 ms (6.6 Hz)? A few more words would help.

L216-221: "In fresh, concentrated smoke plumes, UV active species such as $SO_2$ and aromatic hydrocarbons can give rise to positive artifacts in the $O_3$ absorption measurement (Birks, 2015). … However, such UV-active absorbers are generally not abundant enough in the background atmosphere to be of concern."
This paragraph, which is a consequence of the low selectivity of the method for Ozone in the UV region, leaves a lot of open questions for the reader and is a real weakness of the manuscript. With a detection limit of tens of pptv the average abundance of species with absorption in the UV region, like e.g. BTX, formaldehyde, ketones, … can be in the low ppbv range. The selectivity of the current experimental approach is of clear concern here. In the sentence in line 221 this issue is simply discarded and not enough consideration is given to this issue in this manuscript. The implication is that, if the sample air composition is completely unknown, the interference of other species may render this approach inadequate, if it is applied on its own. A balanced discussion of the selectivity aspects, or potential ways to improve it by combination with other techniques, must be included in the manuscript. Finally, in line 220 the campaign data are said to be "*quality filtered*". If quality filtering means excluding data when formaldehyde mixing ratios were above 5 ppbv, then this is a rather crude way of doing this. Again, there must be a way of also identifying formaldehyde at the same time by other means.

L230: Cross plot  ->  Correlation plot

L232: Data from 15 extra flights could go into the supplementary material.

L253: …not shown in Figure 6.

L257/258: "Scalar data processing included detrending the scalar mixing ratios by subtracting a 20 second running mean and synchronizing the data with the vertical winds." On what basis was that decided. Some more background explanation would be useful here.

Generally, the manuscript would benefit from a brief and compact comparison with other techniques for ozone detection and how the new ROZE compares in performance with those approaches. This could take the form of a small table. If this turns out to be too formidable, the authors may restrict this comparison to cavity enhanced absorption approaches.

***Technical corrections (small improvements and formal/formatting errors):***

Many superscripts (in units) did not come out correctly in print, see lines:
19, 45, 189, 190, 244, 266, 283

L43: Rephrase: … cylinders containing compressed toxic gases and the use of dangerous chemical dyes.

*References:*

Aubinet et al. – relevant pages might be missing
Birks – typo in titles and reference incomplete
Bourgeois et al. –  volume seems to be missing
Ryerson et al. –   NO 2 -> $NO_2$   (insert proper index)
Serdyuchenko et al. – ill-defined symbols in my copy
Young et al. – Journal name should not be abbreviated

*Table:*

Size – unit should read $cm^3$, or   X cm x Y cm x Z cm
Precision – typo in value and unit

*Figures:*

*Figure 1*
• Even though this figure is only schematic, the way the LED light pulses are drawn is confusing. Pulses arrive at cavity with a 10 Hz repetition rate. At 10 Hz only one pulse is in the cavity at one time. The pulse duration is long enough to draw this figure in CW mode and explain the pulsed nature of ROZE in the text.
• Fig. 1 caption – replace *"long optical pathlength"*  by *" long effective optical pathlength"*

*Figure 2*
Axes titles and colour code could be improved. Axis title for the LED spectrum (normalized intensity) is okay, for the cross-section the chosen (right) axis title is also okay. The figure however also contains the reflectivity spectrum, which is unitless, and thus the title should occur somewhere. The relative intensity axis can of course be used, but the title should be changed to make clear that 3 different quantities are shown. Moreover, the two shades of blue are difficult to distinguish, and do not work in my opinion; modify the color code.

*Figure 6*
• O3 on the axis title should be indexed properly; i.e. $O_3$.
• In panel (a) it would be better to show an individual pulse and a proper fit rather than 5 pulses, where the fitted data point can hardly be made out. The sentence: "*Individual pulses were fit to an exponential decay using the selected data points in red*" is not clear. An exponential function is fitted to the experimental data and not vice versa. This should be rephrased.
• What is meant by an "e/3 flush rate"? This is phrased casually.

*Figure 7*
• O3 on both axis titles should be indexed properly; i.e. $O_3$.
• Scatter plot  ->  Correlation plot
• How does an intercept of "0.17 +/- 0.02 ppbv" agree with the detection limit? Comment.

---

## Referee Comment (RC2) · Anonymous Referee #2 · 22 Sep 2020

The paper offers a clear and concise description of a new, sensitive, and versatile instrument for the in situ detection of ozone via UV absorption. The small sample size/rapid flush rate and precision of the instrument enable flux measurements. Comparison/calibration with a reference standard in the laboratory, and comparisons with an established airborne instrument in the field establish its performance characteristics and accuracy. The paper is well written, reasonably comprehensive, and the instrument is a valuable addition to the suite of airborne ozone sensors.

Comments, but Mostly Questions:

1. Line 67 (also caption to Fig. 1) - is the light collimated and coupled via high-

reflectivity mirrors, or do the high-reflectivity mirrors constitute the optical cavity?

2. Lines 75+ and 152+ - It is addressed, but I wonder if it is possible to be a bit more explicit here about what is measured (Iz and I?), what is known (Rayleigh and ozone cross sections?) and what has to be empirically determined/calibrated (Leff, which is set by mirror reflectivity?)?

Also, how stable is mirror reflectivity over time? Do the Iz measurements at different pressures enable determination of Leff in flight?

Are there other atmospheric absorbers in this region?

Does the Rayleigh scatter term depend upon the composition of the sample other than ozone, e.g., $H_2O$, $CO$, $CO_2$? Does the scrubber alter the concentrations of these species?

3. Lines 105+ - Is the cell surface treated to limit ozone loss? What material is the diffuser (FEP?)? Does the particulate filter lead to ozone loss in the sample?

How do you verify the performance of the scrubber? Does its ability to fully scrub ozone depend upon flow rate, ambient pressure, ambient ozone concentrations?

4. Fig. 7 - Any thoughts on what led to the positive offset between ROZE and NOyO3? At <1% of mixing ratios at 20 ppbv it may not be worth worrying about.

---

## Author Comment (AC1) · 19 Oct 2020

**Response to Referee 1**

We would like to thank the Referee for their thoughtful comments. More detailed responses to each point are outlined below.

COMMENTS TO THE AUTHOR(S): This well-written and well-structured manuscript describes the development, validation and application of an instrument based on incoherent broadband cavity-enhanced absorption spectroscopy (IBBCEAS) for the direct detection and quantification of ozone in the UV-C region (~265 nm). The pulsed instrument (Rapid Ozone Experiment, ROZE) is designed for O3 Eddie covariance measurements from airborne platforms and thus requires high time resolution. The authors give an excellent motivation for their work in the introduction. They briefly review the measurement principle before outlining the instruments operation, where sufficient attention to detail is provided to understand the function of the instrument and how the key objectives in instrument performance were met. The lab characterization of the instrument's performance addresses all relevant aspects. Finally, the first application in airborne operation also shows the quality of the instrument for Eddie covariance measurements in comparison with a more established chemiluminescence (reference) approach. The authors pay sufficient attention to detail, error discussion, and relevant critical measurement parameters. The manuscript is basically publishable in its current form, however subject to some improvement and amendments concerning more explanations as outlined below.

- 1. L49: The UV region has the drawback that in this region many other trace species also have significant absorption bands and this puts substantial constraints on the selectivity of the method in that region. See also comment below (L216-220).
- 2. L70-74: The use of an optical filter simplifies the approach and is adequate because the spectrum is unstructured and the absorption has not much or very little "fingerprint character". However, it also illustrates the low selectivity for ozone in this region (see L216-220).

Points 1 and 2 will be addressed in response to question 11 below.

3. L101: The divergence of the LED is quite large – the surface area of LED is also large 1x1 mm^2 (extended light source). Light collection at a distance of 79 mm with a 1" diameter optic causes substantial light loss. Using a beam expander in reverse increases the divergence after partial collimation again by a factor of 2. Can the authors give an estimate what fraction of the overall emission from the LED is actually being imaged onto the cavity – in other words, what is the light collection efficiency for the LED?

We estimate the LED collection efficiency to be at minimum ~1–2%. Additionally, as the mirrors are 99.7% reflective, only ~0.3% of the remaining intensity enters the cavity. Although light losses are large, the PMT detector is not signal-limited. We empirically found that the 79 mm focal length lens and beam expander combination achieved the most effective collimation and optimized the optical pathlength.

4. L103: "...direct the beam 180° into the cell." This sounds strange. Please rephrase or at least refer to Figure 3 here.

The text on L104 has been altered as: ...two mirrors (Thorlabs NB1-K04) turn the beam 180° into the cell (see Figure 3)...

5. L105-114 (section 3.1.2): There is no mention of a purge system. Were the cavity mirrors purged? If not – why not? There is also no mention of an aerosol filter in the inlet at this point, but later the authors point this out (since scattering in the UV is substantial). This section would benefit from mentioning these elements explicitly here.

Purging the cavity mirrors was not found to be necessary to keep them clean. Instead we affixed a filter to the cell inlet port to exclude dust and other particles >  $2\mu$ m. We have edited and moved the appropriate text from Section 3.2 to Section 3.1.2:

L112: A 2-micron pleated mesh filter (Swagelock) affixes to the sample cell inlet port to exclude dust and other particles from affecting the mirror reflectivity, as the mirrors are not independently purged.

The aerosol filter is not a permanent fixture in the instrument; it was added to the flow system as an extra precaution in anticipation of the high aerosol load in smoke plumes and is therefore described in Section 5, "Field demonstration". Although we were still able to achieve an 11 SLM flow rate and acquire eddy covariance  $O_3$  flux measurements, removing the particle filter or replacing it with a higher throughput filter would enable a faster response time for dedicated  $O_3$  flux measurements in a cleaner environment. The text in Section 5 has been modified to read:

L212: The instrument operated as described above, with the addition of an inline particle filter (Balston 9922-05-DQ) to protect the cavity mirrors from fine particulates in the targeted smoke plumes. Although more aggressive filtering comes at the cost of reduced flow rates and thus lowers the instrument response time,  $O_3$  deposition measurements were not a primary objective of FIREX-AQ.

6. L125/126: At this point I was wondering whether there was a flow controller? The flow seems to be only controlled by pump power? How accurate is the flow and how does it vary with pressure variation? (Relevant for aircraft measurements on the ascent and descent). What are the three flows according to the 3 flow speed adjustments?

There is no flow controller and thus the flow varies with pressure: for a fixed ambient pressure, higher sample flows result in lower pressures within the sample cell. The flow also changes in flight with aircraft speed. The pressure gauge provides accurate ( $\pm$  0.2%) pressure readings for calculating the O3 mixing ratio. The flow meter is accurate to within  $\pm$  3%, but this component does not go into calculation of the O3 mixing ratio.

The three flows are user-adjustable and were configured to be roughly 2, 5, and 11 SLM during the FIREX-AQ deployment. These have been noted in the text on L128.

7. L145: Here it would be nice to explicitly learn what on- and off-times a 90% duty cycle refers to? Are there are delay times. Inherently pulsed IBBCEAS is not commonly applied and in the literature the large majority of instruments is indeed continuous wave. Pulsed IBBCEAS can in principle be biased by offsets depending on the measurement timing, which in turn may lead to systematic deviations in the measured absorption signal (see e.g. Keary & Ruth, Opt.

Express, 2019). The authors may want to make an argument that these offsets are not observed here based on the electronics' timing. See also comment on Fig. 1 below.

The duration of the ON and OFF LED pulses (900  $\mu$ s and 100  $\mu$ s respectively) has been added to the text (L143). Due to the fast digitization rate of our ADC unit (100x faster than the LED modulation rate), we have configured the data acquisition to minimize any delay in gated signal averaging, as it was not necessary to introduce any delay for measurement purposes.

8. L182/183: "...and differential scatter or absorption due to non-uniform flow within the sample cell at high flow rates." As in the comment above, more on flow variation and or flow control would be helpful. The instrument might benefit from a pressure controlling flow controller.

See response to question 6 above.

9. L205: "During this experiment, the pump maintained a sample flow rate of 18 SLM." Why was this not done at a lower mass flow of 11 SLM relevant for the airborne measurements with particle filter?

We performed the pulsed value experiment to determine the maximum achievable instrument response time using only the permanent 2  $\mu$ m mesh filter. As described in response to question 5 above, the Balston particle filter could be removed or replaced with a higher throughput filter when using the instrument for dedicated eddy covariance flux observations in environments where we do not expect such a high burden of fine particulates.

10. L206: I am a bit confused here: Please explain "3e fold flush rate". Also compare with caption of Fig 6 which says 1/3e is 9.5 Hz. The 3e-folding time is 150 ms (6.6 Hz)? A few more words would help.

The exponential fit yields a decay constant, or *e*-folding time of tau = 50 ms. The decay is exponential, so the time for O3 to decay to 1/3e (~12%) of its initial concentration is ~105 ms, or 9.5 Hz. The text on L205 has been clarified as: *The time to flush the cell to 1/3e of its initial contents thus corresponds to a flush rate of 9.5 Hz*.

11. L216-221: "In fresh, concentrated smoke plumes, UV active species such as SO2 and aromatic hydrocarbons can give rise to positive artifacts in the O3 absorption measurement (Birks, 2015). ... However, such UV-active absorbers are generally not abundant enough in the background atmosphere to be of concern." This paragraph, which is a consequence of the low selectivity of the method for Ozone in the UV region, leaves a lot of open questions for the reader and is a real weakness of the manuscript. With a detection limit of tens of pptv the average abundance of species with absorption in the UV region, like e.g. BTX, formaldehyde, ketones, ... can be in the low ppbv range. The selectivity of the current experimental approach is of clear concern here. In the sentence in line 221 this issue is simply discarded and not enough consideration is given to this issue in this manuscript. The implication is that, if the sample air composition is completely unknown, the interference of other species may render this approach inadequate, if it is applied on its own. A balanced discussion of the selectivity aspects, or potential ways to improve it by combination with other techniques, must be included in the manuscript. Finally, in line 220 the campaign data

are said to be "quality filtered". If quality filtering means excluding data when formaldehyde mixing ratios were above 5 ppbv, then this is a rather crude way of doing this. Again, there must be a way of also identifying formaldehyde at the same time by other means.

We appreciate the reviewer pointing out that this discussion is not well balanced in the manuscript. The lack of selectivity is a drawback to the UV absorption technique. Given the prevalence of the UV absorption technique in commercial analyzers, much effort has gone into identifying atmospheric species that also absorb in the 265 nm region (see for example the report from 2B Technologies: https://twobtech.com/docs/tech\_notes/TN040.pdf). These include aromatic hydrocarbons and other volatile organic compounds (VOC), which can have absorption cross sections approaching that of  $O_3$ . Therefore, positive artifacts in  $O_3$  can arise in environments with high VOC, such as within smoke plumes (Long et al., 2020) or in polluted urban environments (Spicer et al., 2010).

We have updated the text in the manuscript to present a more balanced and appropriate discussion of the potential for measurement artifacts:

FIREX-AQ flights targeted forest wildfires and agricultural burns. In fresh, concentrated smoke plumes, UV-active species such as  $SO_2$ , aromatic hydrocarbons, and other volatile organic compounds (VOC) can give rise to positive artifacts in the  $O_3$  signal (Long et al., 2020), as the UV absorption technique lacks selectivity (see Birks, 2015). The potential for overestimating  $O_3$  due to interfering absorbers can also be of concern in highly polluted urban environments (e.g., Spicer et al., 2010). In general, these studies demonstrate that UV absorption based  $O_3$  analyzers are not always ideally suited to such applications. Nonetheless, modifications such using an  $O_3$  selective scrubber material (e.g., heated graphite) to preserve VOC and thus account for interferences in the background ( $I_Z$ ) signal have been shown to reduce positive artifacts (Turnipseed et al., 2017). As we did not substitute the ROZE scrubber for the FIREX-AQ deployment, an on-board, independent measurement of formaldehyde (HCHO) was used as a plume indicator. ROZE  $O_3$  data are therefore quality filtered to remove points sampled within dense smoke plumes using HCHO mixing ratios above 5 ppbv.

12. L230: Cross plot -> Correlation plot

This has been corrected in the text.

13. L232: Data from 15 extra flights could go into the supplementary material.

The remaining ROZE and NOyO3 O3 data from FIREX-AQ are publicly available ( https://www-air.larc.nasa.gov/missions/firex-aq/), and the authors feel that providing additional correlation plots does not further the comparison. We have added the range of observed intercepts to the range of observed slopes (text L238).

14. L253: ...not shown in Figure 6.

See response to question 9 above.

15. L257/258: "Scalar data processing included detrending the scalar mixing ratios by subtracting a 20 second running mean and synchronizing the data with the vertical winds." On what basis was that decided. Some more background explanation would be useful here.

The detrend removes any non-turbulent variability in the scalar data. The averaging window was determined empirically by inspection of the co-spectral power (as in Figure 8c) generated from a range of detrending lengths. This data is not shown, but the procedure has been detailed in the text as follows:

L264: Scalar data were detrended by subtracting a 20 second running mean, which corresponds to spatial scales of ~2.7 km. The detrending length was chosen to remove non-turbulent variability (e.g., changing chemical conditions) while still capturing the largest flux-contributing eddies as identified by examination of the co-spectra from a range of averaging windows. Scalar data were then synchronized to the vertical winds using a time-lag that optimized covariance.

16. Generally, the manuscript would benefit from a brief and compact comparison with other techniques for ozone detection and how the new ROZE compares in performance with those approaches. This could take the form of a small table. If this turns out to be too formidable, the authors may restrict this comparison to cavity enhanced absorption approaches.

We appreciate the reviewer's desire for context to be provided. The manuscript briefly describes other techniques in the introduction and provides references without detailing their performance characteristics. Our purpose in this manuscript is to demonstrate the capabilities of ROZE, in particular for eddy covariance fluxes, rather than to compare/contrast it to existing technologies. A more detailed comparison is best addressed in a devoted instrument intercomparison or review article, as instrument drawbacks and benefits also depend on the measurement application (cost, weight and size, precision requirements, accuracy requirements, time response, etc) and deserve a more thorough examination.

Technical corrections (small improvements and formal/formatting errors): The following have been addressed in the text:

Many superscripts (in units) did not come out correctly in print, see lines: 19, 45, 189, 190, 244, 266, 283

L43: Rephrase: ... cylinders containing compressed toxic gases and the use of dangerous chemical dyes.

**References:**

Aubinet et al. – relevant pages might be missing Birks – typo in titles and reference incomplete Bourgeois et al. – volume seems to be missing Ryerson et al. – NO 2 -> NO2 (insert proper index) Serdyuchenko et al. – ill-defined symbols in my copy Young et al. – Journal name should not be abbreviated

Table:

Size – unit should read cm3, or X cm x Y cm x Z cm Precision – typo in value and unit

**Figures:**

Figure 1

- Even though this figure is only schematic, the way the LED light pulses are drawn is confusing. Pulses arrive at cavity with a 10 Hz repetition rate. At 10 Hz only one pulse is in the cavity at one time. The pulse duration is long enough to draw this figure in CW mode and explain the pulsed nature of ROZE in the text.
- Fig. 1 caption replace *"long optical pathlength"* by *"long* effective optical pathlength"

**Figure 2**

Axes titles and colour code could be improved. Axis title for the LED spectrum (normalized intensity) is okay, for the cross-section the chosen (right) axis title is also okay. The figure however also contains the reflectivity spectrum, which is unitless, and thus the title should occur somewhere. The relative intensity axis can of course be used, but the title should be changed to make clear that 3 different quantities are shown. Moreover, the two shades of blue are difficult to distinguish, and do not work in my opinion; modify the color code.

**Figure 6**

- O3 on the axis title should be indexed properly; i.e. O3.
- In panel (a) it would be better to show an individual pulse and a proper fit rather than 5 pulses, where the fitted data point can hardly be made out. The sentence: "Individual pulses were fit to an exponential decay using the selected data points in red" is not clear. An exponential function is fitted to the experimental data and not vice versa. This should be rephrased.
- What is meant by an "e/3 flush rate"? This is phrased casually. See response to question 10 above.

**Figure 7**

- O3 on both axis titles should be indexed properly; i.e. O3.
- Scatter plot -> Correlation plot
- How does an intercept of "0.17 +/- 0.02 ppbv" agree with the detection limit? Comment. See responses to questions 13 and 20.

**Response to Referee 2**

We would like to thank the Referee for their thoughtful comments. More detailed responses to the comments are outlined below.

COMMENTS TO THE AUTHOR(S): The paper offers a clear and concise description of a new, sensitive, and versatile instrument for the in-situ detection of ozone via UV absorption. The small sample size/rapid flush rate and precision of the instrument enable flux measurements. Comparison/calibration with a reference standard in the laboratory, and comparisons with an established airborne instrument in the field establish its performance characteristics and accuracy. The paper is well written, reasonably comprehensive, and the instrument is a valuable addition to the suite of airborne ozone sensors. Comments, but Mostly Questions:

17. Line 67 (also caption to Fig. 1) - is the light collimated and coupled via high-reflectivity mirrors, or do the high-reflectivity mirrors constitute the optical cavity?

The high-reflectivity mirrors constitute the optical cavity. The beam is collimated using an aspheric lens. The text and figure caption have been clarified accordingly:

L67: As illustrated in Figure 1, a light-emitting diode (LED) in the UV ( $\lambda_{max}$  = 265 nm) is collimated with an aspheric lens and coupled into an optical cavity formed by two high-reflectivity mirrors.

18. Lines 75+ and 152+ - It is addressed, but I wonder if it is possible to be a bit more explicit here about what is measured (Iz and I?), what is known (Rayleigh and ozone cross sections?) and what has to be empirically determined/calibrated (Leff, which is set by mirror reflectivity?)? Also, how stable is mirror reflectivity over time? Do the Iz measurements at different pressures enable determination of Leff in flight? Are there other atmospheric absorbers in this region? Does the Rayleigh scatter term depend upon the composition of the sample other than ozone, e.g., H2O, CO, CO2? Does the scrubber alter the concentrations of these species?

The intensity terms ( $I_z$ , I) are measured, the Rayleigh and ozone cross sections are known from the literature. The mirror reflectivity dictates the cavity excitinction ( $a_{cav}$ )/effective pathlength ( $L_{eff}$ ) and must be empirically determined via calibration. The text has been clarified as follows:

L85: In principle, accurate trace gas measurements require calibration of the  $\alpha_{cav}$  term yielding  $L_{eff}$ , knowledge of the Rayleigh and absorption cross sections in the detected spectral region, and the measured  $I_0$  and I terms.

L155: Using the measured  $I_Z$ , I, and the known Rayleigh scatter and  $O_3$  absorption cross sections, the  $O_3$  number density can then be determined as  $\alpha_{O3} = N_{O3}\sigma_{O3}$ .

With adequate particle filtering, the mirror reflectivity is relatively stable over time. Although we do not show the data in the manuscript, calibrations before and after the FIREX-AQ field deployment do not show significant degradation in the optical pathlength. Changes in Iz with pressure during flight could enable real-time calibration. Other atmospheric absorbers are present in the region. A more detailed discussion of this point can be found in response to question 11 above.

We use the Rayleigh scattering cross section for the standard atmosphere (dry air, 300 ppmv CO2) as determined by Bucholtz (1995). Given the relatively low ambient mixing ratios of H2O, CO2 and CO, the scattering is dominated by interactions with N2 and O2. Tomasi et al. (2005) confirm that increasing to ~50% relative humidity and 385 ppmv CO2 results in a < 0.02% difference in the calculated cross section. Overall, ambient changes in these species contributes only small uncertainty and is lumped into the upper estimate of ± 3% in the Rayleigh cross-section. The Carulite scrubber does not destroy CO2 or H2O but does convert CO into CO2.

19. Lines 105+ - Is the cell surface treated to limit ozone loss? What material is the diffuser (FEP?)? Does the particulate filter lead to ozone loss in the sample? How do you verify the performance of the scrubber? Does its ability to fully scrub ozone depend upon flow rate, ambient pressure, ambient ozone concentrations?

The cell surface is not treated to limit  $O_3$  loss. We have found no evidence of surface  $O_3$  loss during regular operation, and the Rayleigh and  $O_3$  calibrations show good agreement. The diffuser is made of aluminum, which has been added to the text.

We verify the performance of the  $O_3$  scrubber by configuring high  $O_3$  (~ 1 ppmv) and zero  $O_3$  air to flow through it. We find that the resulting detected intensity shows no change with increasing  $O_3$  under both low and high flow conditions.

20. Fig. 7 - Any thoughts on what led to the positive offset between ROZE and NOyO3? At <1% of mixing ratios at 20 ppbv it may not be worth worrying about.

ROZE and NOyO3 did not share a sample line during the FIREX-AQ deployment. Although we cannot conclusively identify the source of the non-zero intercept, NOyO3 sampled from an inlet at the left forebody of the aircraft while ROZE sampled from an inlet on the right, midbody. Perhaps small fluctuations in ambient O3 could account for some of the observed discrepancy. We have modified the text as follows:

L257: Note the intercept is less than 1% of the minimum observed  $O_3$  mixing ratios for this flight. Comparisons for 15 flights from the campaign indicate a range of 0.96–1.04 in slope and -1.6–1.4 ppbv  $O_3$  in intercept (in all cases, this offset is < 4% of the minimum measured  $O_3$ ), consistent with the measurement uncertainty.

---

## Author Response (AR2)

Response to the Editor

We would like to thank the Editor for the helpful comments on the manuscript. Responses to each point are detailed below.

COMMENTS TO THE AUTHOR(S): The reviewers have recommended your manuscript for publication and have emphasized on its high quality. Moreover, you have responded to all issues that have been raised. I am therefore very happy to decide that the manuscript can be published after minor revisions. Please find the list of minor requests here below.

1. Write "1/(3e) folding time" or "1/(3e)" flush rate instead of "1/3e folding time" etc., which has created some confusion throughout the document. I wonder whether 1/(3e) is commonly used terminology? Is there a particular reason for not using the 1/e^2 or the 10 % levels instead?

   Although the instrument response time is often characterized by the e-folding time, we use the 3e-fold definition to translate the response time into a flush rate that estimates how fast the instrument acquires independent samples. We have updated all the terminology from 1/(3e) to read 3e-fold(ing) for simplicity and clarity.

2. Fig. 4 and bottom of page 5: please indicate the confidence level of uncertainty values.

   The changes suggested by Points 2-8 are highlighted in the revised text.

3. Top of page 6: "1 sigma Allan plot" is unconventional terminology. Use "Allan-Werle plot" instead.

4. End of section 4.2: use "conservative uncertainty" instead of "upper uncertainty"

5. line 104: "response times" instead of "a response times"

6. line 127: "standard liter per minute" instead of "standard liters per minute"

7. line 278 "shows" instead of "show"

8. line 294: The phrase around "under sampling" is difficult to understand. Do you mean "undersampling"?

   Yes, the text has been modified accordingly.

9. Fig. 1: The axis scale for the ozone cross section is not well defined. Please add markers for the 0.0 and the 1.0e-17 cm^2 levels. One has the impression that the peak value is at 1.2e-17 cm^2 instead of 1.15e-17 cm^2 given in the Gorshelev et al. (Atmos. Meas. Tech., 7, 609–624, 2014) paper. The figure could also be improved by adding the sensitivity weighted cross section, ie the product LED*R*sigma.

   We have updated the plot axes to make the cross section values easier to identify.

[revised manuscript text omitted]